# The Validity of the Push Band 2.0 on the Reactive Strength Index Assessment in Drop Jump

**DOI:** 10.3390/s22134724

**Published:** 2022-06-22

**Authors:** Raynier Montoro-Bombú, Lázaro de la Paz Arencibia, Carlo Buzzichelli, Paulo Miranda-Oliveira, Orlando Fernandes, Amândio Santos, Luis Rama

**Affiliations:** 1Faculty of Sport Sciences and Physical Education, University of Coimbra, 3004-531 Coimbra, Portugal; acupidosantos@gmail.com (A.S.); luisrama@fcdef.uc.pt (L.R.); 2Faculty of Sports, University of Physical Culture and Sport Sciences Manuel Fajardo, Havana 10600, Cuba; lazarodpaz50@gmail.com; 3Faculty of Exercise and Sport Science, University of Milano, 20122 Milan, Italy; cb@isci.education; 4Interdisciplinary Research Centre Egas Moniz (CiiEM), Cooperativa de Ensino Superior CRL, 2829-511 Almada, Portugal; paoliveira@egasmoniz.edu.pt; 5School of Technology and Management (ESTG), Polytechnic of Leiria, 2411-901 Leiria, Portugal; 6Portuguese Athletics Federation (FPA), 2799-538 Oeiras, Portugal; 7Sport and Health Department, School of Health and Human Development, University of Evora, 7000-671 Evora, Portugal; orlandoj@uevora.pt; 8Comprehensive Health Research Center (CHRC), University of Evora, 7000-671 Evora, Portugal; 9Research Unit for Sport and Physical Activity (CIDAF), 3004-531 Coimbra, Portugal

**Keywords:** reactivity assessment, plyometrics, drop jump, instrument, vertical jump

## Abstract

This study aims to verify the validity of the Push Band 2.0 (PB_2.0_) device on the reactive strength index (RSI) measurement, using a force plate (FP) and an optical sensor device, OptoJump (OPT), as a reference. Twenty trained athletes performed 60 drop jump trials with a height box of 30 cm. A randomized repeated measures study was conducted during a single session using the PB_2.0_, the OPT, and the plate force manually synchronized to obtain RSI data for each jump. Validity was analyzed by contrasting three measures: the intra-class correlation coefficient (ICC), the Bland–Altman test, and R2 coefficient of determination. Bland–Altman analysis showed that RSI and FP for PB2.0 (media = −0.047; IC 93.34%) of all data were within the confidence interval, indicating a statistically reliable result. The RSI measured by the OPT and PB2.0 also provided similar values (media = −0.047). These data are identical to other validity measures (ICC and linear correlation) but differ in the R2 values. The explained variation of PB2.0 measures attained only 29.3% of the FP (R2 = 0.293) and 29.5% (R2 = 0.295) of the OPT assessment, showing a very low determination coefficient. The results of this study point to caution in the use of PB2.0 when measuring RSI in scientific research.

## 1. Introduction

Plyometric exercises are commonly considered in training programs to improve reactive strength [1,2,3]. The drop jump (DJ) is one of the most studied protocols in the plyometric assessment [4,5,6], and some studies have demonstrated its utility to assess the influence of leg extensor strength qualities on the vertical jump performance [7,8,9,10,11].

Reactive strength index (RSI) is an effective marker of reactive strength [12] due to the use of a fast shortening caused by a previous activation in the DJ, known as the stretch–shortening cycle [13,14]. The duration of the ground contact time (GCT) characterizes two types of stretch–shortening cycle: short duration if >250 ms, as it is typical of the DJ, and long duration if >250 ms is present in the counter movement jump (CMJ). The RSI is currently one of the broadest performance markers reported in the literature [13,15,16] and it is among the most frequently assessed, representing a measure of reactive strength [16]. This index was proposed as a consistent and valid indicator to control the stress produced in the tendon muscle complex during jumps that incorporate a drop phase [17], highlighting the capacity when moving from an eccentric to a concentric contraction [18].

Some research [15,16,17,19,20] considers the RSI to be a practical way to evaluate the quality of jump performance using only a contact mat [20,21] or a force plate [13]. In addition, research has reported its use to establish recommendations for optimizing the jump from different heights in the plyometric training [15] and even to evaluate neuromuscular fatigue after training sessions [22,23]. It should be noted that during DJ execution, RSI equals the ratio of the jump height (HJ) and the ground contact time (GCT) [24]. 

The DJ could be one of the few plyometric exercises that are commonly used to analyse bilateral rebound mechanics [25] and which contains GCT.

Recently, an accelerometer-based device, the Push Band 2.0 (PB2.0) (Push now partner WHOOP, Toronto, ON, Canada), provided, among other measures, the jump height and the RSI in the DJ performance. In addition to its relatively affordable cost, if confirmed, the validity of this device would present numerous advantages for coaches as they spend several weeks away from their athletes throughout the year-round athletic preparation determined by the sports elite level. This device during the jump training would allow RSI metrics to be observed in real time through the integrated web-based data transmission system. In addition, during exercising tasks in the training, it is difficult to predict with the naked eye when the athlete has a slight decrease in reactive strength, a parameter that this device can detect, indicating to stop the DJ’s training immediately. The PB2.0 has previously been validated for speed-based strength work [26] and squat jumps [27]. It has also shown good correlations with different jumps, similar to other applications and devices [28]. It seems counterproductive that the latter standardizes a 90-degree angle for all jumps, which could affect the RSI by increasing contact time, contradicting previous findings [20,29], and could affect the reliability of RSI validation. The authors also report possible proceeding errors due to the possibility of uncontrolled movements of the PB2.0 belt during the jumps.

In addition, an optical measurement system called OptoJump (OPT) has also been previously validated [30] with excellent test–retest reliability and ICCs ranging from 0.982 to 0.989, with low coefficients of variation (2.7%) [28,31,32,33]. This equipment allows real-time evaluation, increasing the possibilities for decision-making in plyometric jump training. The present study aims to evaluate the validity of the PB2.0 device for RSI assessment in the drop jump, using a force plate (FP) and OPT as standard references.

## 2. Materials and Methods

### 2.1. The Experimental Approach to the Problem

A randomized repeated measures study in a single session was conducted to test the hypothesis of the validity of PB2.0 in the assessing of RSI during the performance of drop jump from 30 cm height. 

### 2.2. Subjects

Twenty-two athletes were recruited for this study. Of these, 17 were volleyball athletes (12 men and 5 women), and 3 were track and field triple jump specialists (3 men) (mean ± SD; age: 20.75 ± 1.67 (year), height: 1.74 ± 0.06 (m), weight 64.76 ± 9.67 (kg), and BMI 20.52 ± 2.93). The sample characteristics are presented in Table 1. 

The adopted inclusion criteria were: (1) more than two years of regular training with competitive participation; (2) previous experience in plyometrics training; (3) absence of significant pathological or past traumatic events in the lower limbs; (4) no present flat feet condition. All participants were informed about the experimental procedures, possible harms, and benefits of the study, and gave written consent. The research was conducted following the recommendations of the Helsinki declaration (2013) and was approved by the scientific board and the ethics committee of the Faculty of Sports Sciences and Physical Education of Coimbra University.

### 2.3. Procedures

An indoor sports venue was used for the data collection session. Each athlete performed a 10 min general warm-up routine which consisted of running at low intensity and 5 min of active recovery. Then, the specific warm-up was followed by a muscle memory activation task [34] with specific plyometric jumping exercises that consisted of 3 countermovement jumps with 2 min recovery and 3 drop jump attempts at 30 cm height (DJ30). Then, a 5 min passive recovery period followed. Before performing the tests, the subjects’ height, body mass, and age were collected.

A 30 cm box height was used, and each athlete performed 3 drop jump attempts (DJ30) following previously established norms [5,20,35,36]. To reduce possible errors in the RSI measurement and reduce the effects of PB_2.0_ movement during jumping [28], the belt was fixed on the subjects with double-sided tape between L3 and L4 (Figure 1). PB_2.0_ was also set inside the belt. The DJ was performed with the hands placed on the waist, and the first starting leg was the right leg. The jump was invalidated if the landing occurred with only one leg. Subjects and their coaches were instructed to avoid intense exercise 24 h before the evaluation session.

The measurements were performed using a manual synchronization of the PB_2.0_, OPT, and an FP (Kistler Model 9260AA6, Winterthur, Switzerland). The protocol was applied in a week of overall low training load, assuming no plyometric sessions in the previous 5 days.

### 2.4. Instruments

In the validation of the PB_2.0_, the OPT and the FP were manually synchronized just before the jump attempts. The FP was placed on a flat and compact surface. The 30 cm jump box was placed 10 cm from the rear edge of the FP. The OPT was located at both sides of the FP, one meter apart between the optical emitter and the receiver. The level of the OPT was adjusted to 2 mm above the FP contact surface, as was proposed by the manufacturers. The synchronization strategy made it possible to simultaneously record the data provided in each jump attempt in both types of equipment, which is fundamental for validation proposals.

Push Band 2.0 is a device that works with PUSHCore, Toronto, ON, Canada (Figure 2), and it is the latest algorithm for using the triaxial accelerometer to provide peak and average velocity data, with a full range of ±16 g and a sensitivity of 2048 least significant bits/g. It incorporates a 3D gyroscope with a full range of ±2000 degrees/s and a sensitivity of 16.4 bits. It measures instantaneous vertical velocity with a sampling rate of 1000 Hz. It has a rechargeable lithium polymer battery with a charge cycle of 1.5 h with a power consumption of 50 mA during charging. It is 77.5 mm high, 55.3 mm wide, 15 mm deep, and weighs 32 g. This equipment is compatible with iOS 12 and above or Android 7.0 and above using Bluetooth 4.2 and above transmission protocol. It was installed according to the brand’s specifications, and collected data were recorded on a Lenovo yoga tablet and Android 9 operating system. PB2.0 records the height of the jumps using the following previously published equation:Jump height = 9.81/8 × FLIGHT TIME ^2^(1)

The RSI is calculated using Equation (2) expressed in meters per second [16]:RSI = HJ (cm)/GCT (s)(2)

HJ represents the height of the jump and GCT is the ground contact time during the landing. The force plate (60 × 40 cm) was the reference for validating the data provided from PB_2.0,_ and OPT was placed on a compact surface to reduce noise during the jumps. It was calibrated according to previous studies [4,37]. Data were collected and displayed in real time at a sampling rate of 1000 Hz using an interface box (Kistler Model 9260AA6, Winterthur, Switzerland), and were analyzed using Bioware 5.3.2.9 software (Winterthur, Switzerland) following the manufacturer’s instructions.

The optical contact measurement system OPT—nexX30 (Bolzano, Italia) [30,38] was adjusted according to the abovementioned standards. Jump-related data were extracted from the software produced by the company in version 1.12.19.0. Anthropometric assessment: Height was measured with a stadiometer to within 0.1 cm (Bodymeter 206, SECA, Hamburg, Germany). Body mass was assessed with the force platform, and the body mass index was evaluated, considering previous protocols [39].

**Figure 2 sensors-22-04724-f002:**
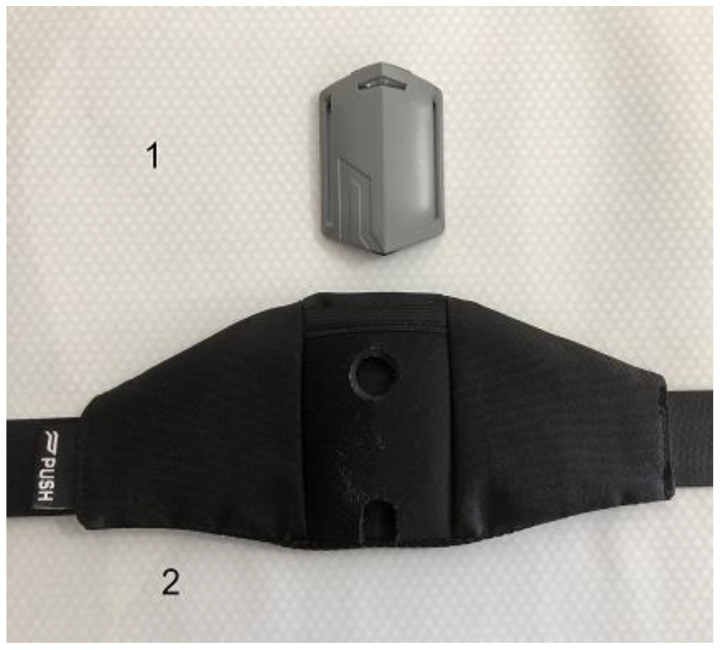
Sample 1: the Push Band 2.0 mobile inertial unit; Sample 2: the plyometric jumping belt supplied by the company.

### 2.5. Statistical Analysis

Descriptive statistics were used to obtain the means, standard deviations, and confidence intervals. Assumptions of normality were checked with the Shapiro–Wilk test. The statistical power of this study of the difference between two dependent means (pairwise) reached a beta value of 86.2%, with an alpha of 0.05 and a moderate effect size of 0.6, and demanding a sample size of 21 subjects. The validity calculation was obtained by comparing two measures, applying the following strategy: first, the intraclass correlation coefficient (ICC) examined the agreement between the instruments. This was adjusted in the mixed factor model and type: absolute agreement, with a 95% confidence interval. The second method was the Bland–Altman test. A one-sample t-test was used to compare whether the measures were statistically different from zero. Finally, a linear regression model was performed to verify the explained variation. In all analyses, the significance provided was 5% (*p* < 0.05). Data analysis was performed with the statistical program SPSS, V.27.0 and the graphs were produced with the statistical software GraphPad Prism Version.9.4.0 (San Diego, CA, USA).

## 3. Results

We conducted a comparative *t*-test analysis of the mean difference for the 60 trials (all data were collected with manual synchronized equipment OPT, FP, and PB_2.0_). No differences were found between RSI assessments: PB2.0 vs. the FP (*p* < 0.389); PB2.0 vs. OPT (*p* < 0.400). 

As expected, the data from OPT vs. FP show identical results for RSI (*p* < 0.701) and HJ (*p* < 0.569). These values showed high concordance between the methods (Table 2).

Figure 3a shows that the mean difference RSI values between FP and PB2.0 were equal to −0.047, with the mean being close to 0 and 93.34% of the data within the confidence interval. Figure 3b shows the mean difference values of RSI of OPT and PB2.0; the mean values were (0.046), close to 0, with 93.34% of the data within the confidence interval. The data show that PB2.0, according to this statistical method, showed reliable values.

Figure 3c shows the mean difference of the RSI between FP and OPT, and Figure 3d the mean difference of the HJ between FP and OPT, which underwent expected results. These later data are almost perfect (Figure 3c = 0.001, Figure 3d = 0.000), with the confidence intervals within accepted standards. 

The linear regression analysis confirms no systematic errors between the methods. Data showed no tendency to overestimate or underestimate, and had a homogeneous trend since they had significant values (*p* < 0.005) and an acceptable range of error. In Figure 4a, RSI values between FP and PB2.0 (*p* < 0.691) are shown, while in Figure 4b, RSI values between OPT and PB2.0 (*p* < 0.677) are shown, and the standard error for these two methods was 0.143. RSI and HJ values between FP and OPT show statistically near-perfect-significance values and standard errors (Table 3).

The Figure 4a reports the RSI of FP versus PB2.0, showing high acceptable agreement (ICC = 0.703). The same was found with the RSI of OPT versus PB2.0 (ICC = 0.704) in (Figure 4b). These values of ICC confirm the dispersion of the data already found with Bland–Altman analysis, with the criterion of 1.96 SD, in addition to the standard error (0.143) shown in the linear regression. The RSI between FP and OPT and HJ between FP and OPT, in the Figure 4c,d, respectively, are statistically perfect (ICC = 0.999; ICC = 1.00). 

The determination coefficient (R^2^) was used to determine the explained variation of PB_2.0_ from the FP and the OPT (Figure 5). Results showed that the variation in PB2.0 measures could be reproduced at R^2^ = 0.252, which represents a true and lower value (Figure 5a). These data are also evident in the relationship between PB_2.0_ and OPT (R^2^ = 0.254) (Figure 5b). These R^2^ results could already be expected due to the tendency for data dispersion shown by BP_2.0_ in the linear regression, Bland–Altman, and ICC analyses. Finally, this research also corroborates almost perfect expected results for RSI and HJ between FP and OPT (R^2^ = 0.999–0.998, respectively) (Figure 5c,d). 

## 4. Discussion

The present study aimed to evaluate the validity of the PB2.0 device for assessing RSI in drop jumping, using an FP and an OPT as a standard reference. The main results of the comparative analysis for independent samples showed that the RSI results (PB2.0 vs. FP) and (PB2.0 vs. OPT) are not statistically different (*p* < 0.389 and *p* < 0.400, respectively). The standard error of measurement in the PB2.0 is higher than that found in the reference instruments. These findings show that although there is an agreement between these measuring instruments, they tend to have a higher dispersion of data. On the other hand, the RSI and HJ results (OPT vs. FP) show excellent agreement and a very low standard error found previously [28,38].

The Bland–Altman analysis of RSI between PB2.0 against FP showed that the confidence intervals were close to zero (−0.047); the same was found with the RSI of PB2.0 against OPT (−0.046). The hypothesis that PB2.0 is a reliable instrument to measure RSI concerning FP is accepted. Although PB2.0, in general, satisfied the statistically demand of validity, in our opinion, it is crucial to analyze the constricting of the SD values (±1.96).

In the Bland–Altman analysis, the SD of RSI with PB2.0 against FP equals 0.788–0.884, and for PB2.0 against FP, 0.882–0.789. These SD results reinforce our opinion to maintain a reserved criterion, similar to the comparative t-test, where a tendency for PB0.2 to scatter the data was observed. 

A previous study [28] considered that the possible source of measurement errors with the PB2.0 could be due to possible movements during the jump. These problems have been corrected (see Section 2.3), but data dispersion during the DJ30 remains. Previous research [26] recommends that this device offers the best results in slow movements, so it may not accurately detect the explosive velocity variations produced during drop jumps. Although the PB2.0 may be a valid tool for assessing velocity in strength work [26,40], we should consider it when using it to monitor the drop jump. Although the ICC is considered suitable for assessing the devices’ validity [27,28,41], our data showed lower correlation values of the ICC for the RSI measured with PB2.0 and FP, indicating only an acceptable validity. Accordingly, our results contradict, in part, others [27,28,41] which showed higher ICC values, although not recognizing that (ICC) has no units, which makes it challenging to translate the impact of the variability in real practice. Moreover, statistically, the ICC can show a very high correlation because it reports that as one value grows, the other also tends to increase, or the contrary. Two different instruments that coincide only slightly can also have high correlation indices.

The coefficient of determination (R^2^) indicates the strength of the association between the variables and has been used in several instrument validation and reliability studies [26,27,28]. Our results showed lower explanation values than previous studies [27,28]. 

Although the present study presents promising results, it has some limitations since we evaluate the RSI in the DJ from a 30 cm height box. It would be interesting to assess with other heights. In addition, it would be essential to clarify the PB2.0 procedure to determine the velocity in the DJ and thus be able to perform validations compared to the force plate. These parameters were not analyzed in previous studies [40]. In addition, the same studies report that they do not show accurate data for the average velocity. 

The applications of this study are novel for coaches measuring RSI from the metrics reported by the BP2.0. The study demonstrates that this device, while valid, does not express RSI values with the same validity as FP and OPT.

## 5. Conclusions

Our results show that the PB2.0 is a valid device for measuring RSI in sports training. However, strength and fitness professionals should consider the results of this research before using the PB2.0 to assess RSI. The RSI with the OPT equipment was proven to be valid and reliable, as the values of its HJ calculation show acceptable validity. Finally, when comparing with previous research, despite finding some differences in the results, they warn about the unreliability of the PB2.0 device for use in scientific research.

## Figures and Tables

**Figure 1 sensors-22-04724-f001:**
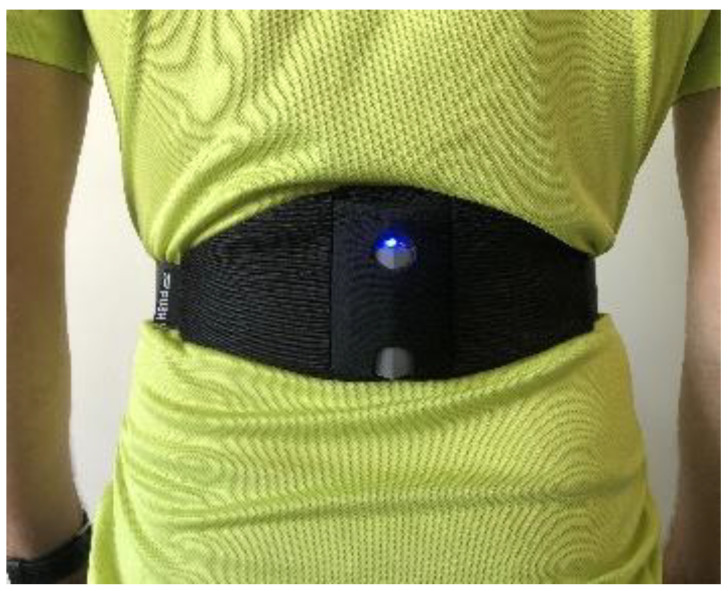
Example of belt location with double-sided adhesive tape between L3 and L4.

**Figure 3 sensors-22-04724-f003:**
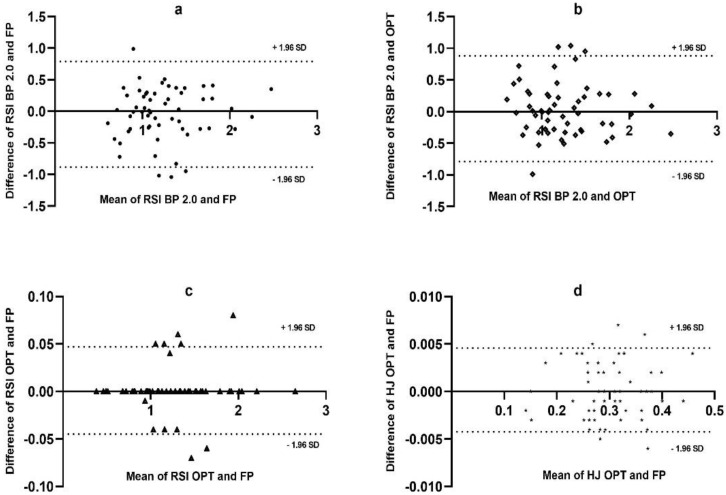
Agreement of the results of the Bland–Altman analysis. The *x*-axis represents the values of the means, and the *y*-axis represents the differences between the means. The criteria for all analyses were (+1.96 SD) as the upper range and (−1.96 SD) as the lower range. (**a**) Bland–Altman analysis RSI between PB2.0 and FP; (**b**) Bland–Altman analysis RSI between PB2.0 and OPT; (**c**) Bland–Altman analysis RSI between FP and OPT; (**d**) Bland–Altman analysis HJ between PB2.0 and OPT.

**Figure 4 sensors-22-04724-f004:**
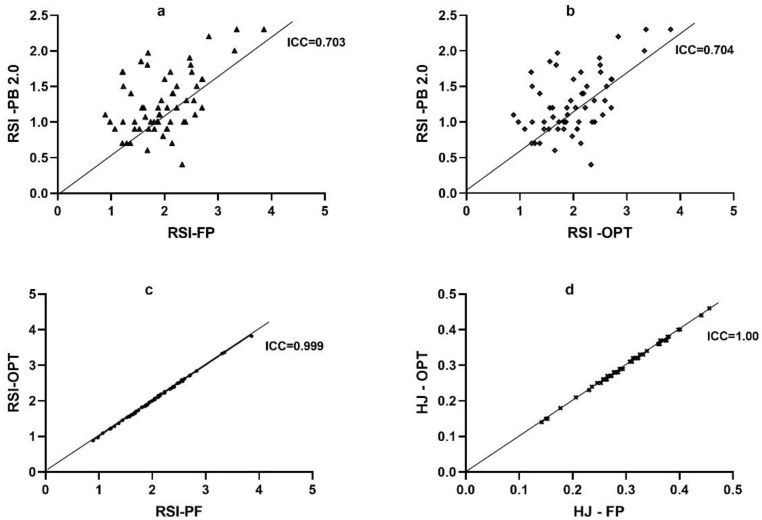
Agreement between the used methods. (**a**) intraclass correlation coefficient of RSI between PB2.0 and FP; (**b**) intraclass correlation coefficient of RSI between PB2.0 and OPT; (**c**) intraclass correlation coefficient of RSI between FP and OPT; (**d**) intraclass correlation coefficient of HJ between PB2.0 and OPT.

**Figure 5 sensors-22-04724-f005:**
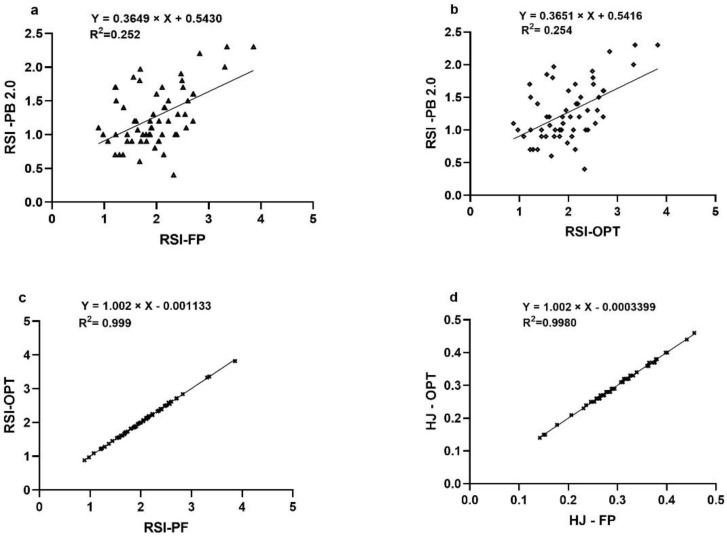
The R2 correlation coefficient analysis. (**a**) The R2 of RSI between PB2.0 and FP; (**b**) The R2 of RSI between PB2.0 and OPT; (**c**) The R2 of RSI between FP and OPT; (**d**) The R2 of HJ between PB2.0 and OPT.

**Table 1 sensors-22-04724-t001:** Anthropometric characteristics of the subjects.

	Men (*n* = 12)	Women (*n* = 8)	Total (*n* = 20)
Age (year)	21.00 ± 1.71	21.00 ± 1.57	21.00 ± 1.64
Age group (year)	18–24	18–23	18–24
Weight (kg)	65.20 ± 8.68	62.92 ± 8.17	64.06 ± 8.43
Height (cm)	178 ± 6.0	167 ± 4.0	172 ± 5.0
BMI (kg·m^2^)	19.52 ± 2.63	21.77 ± 3.10	20.64 ± 2.87

**Table 2 sensors-22-04724-t002:** Concurrent validation t-test significance and Bland–Altman between measuring devices during DJ30.

	Sig.	Bland–Altman (1.96 SD)
RSI FP vs. PB_2.0_	0.389	−0.047 (0.788–(−0.884))
RSI OPT vs. PB_2.0_	0.400	−0.046 (0.882–(−0.789))
RSI OPT vs. FP	0.701	0.001 (0.047–(−0.044))
HJ FP vs. OPT	0.569	0.000 (0.004–(−0.004))

Reactive strength index (RSI); force plate (FP); OptoJump (OPT); Push Band 2.0 (PB2.0); jump height (HJ).

**Table 3 sensors-22-04724-t003:** Concurrent validation of linear regression and ICC between measuring devices during DJ30.

	Regression Lin		
	Desv. Error	Sig. (95% CI)	CCI (95% CI)
RIS FP vs. PB_2.0_	0.143	0.691	0.703 (0.503–0.822)
RSI OPT vs. PB_2.0_	0.143	0.677	0.704 (0.505–0.823)
RSI OPT vs. FP	0.007	0.761	0.999 (0.999–1.00)
HJ FP vs. OPT	0.006	0.691	1.00 (0.999–1.00)

Reactive strength index (RSI); force plate (FP); OptoJump (OPT); Push Band 2.0 (PB2.0); jump height (HJ).

## Data Availability

Not applicable.

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
