# Peer review of "The Validity of the Push Band 2.0 on the Reactive Strength Index Assessment in Drop Jump"

_sensors, 2022, doi:10.3390/s22134724_

Round 1
Reviewer 1 Report
The paper validates a device (Push Band 2.0) during the execution of the drop jump movement to measure the reactive strength index. The authors achieved satisfactory results, demonstrating that the device may be able to provide this analysis with a particular variation. Some points must be analyzed.
Major
- Check the possibility of increasing the number of tests to validate the equipment. Why was only the DJ chosen with a 30 cm box? In addition, will the device provide the same results for other highs?
- It is necessary to insert a figure with a photo of the devices, further detailing the experimental protocol.
- Explain the exclusion of flat feet subjects.
- A more detailed description of the equipment should be performed, especially for Push 2.0.
- The equations are not numbered. Explain the principle of equation 1. Where was it based?
- The authors did not mention with more specificity the advantages of the use PB2.0. It needs to be highlighted.
- Figures 1, 2, and 3 have poor quality and two legends. Please, correct them.
Minor
- Revise the name of the authors.
- Please insert a space between the number and the unities.
- “Some research [15-17, 19, 20] considers”. If there is more than one author, please reconsider the sentence.
- Explain the terms after the abstract, such as PF, which is only explained in the abstract.
- “(…) taking, into account (…)”, please, rewrite.
- The paragraphs that mention the discussion about Fig. 1 present (A), (B) … (D) to refer to Fig 1a, b, … d. Please, revise all the text to maintain the order in the presentation of the text.
- Table 1 title is separated from the table.
- “Research [27,28,41] show”, please review the sentence.
Author Response
Response to Reviewer 1 Comments.
We thank the reviewer for the comments. Your appropriate comments and contributions allow us to improve our paper. The authors appreciate the value of the comments and recognise that they were important in consolidating the document. They will also try to respond to each question based on a review of the content and supported by the previously published bibliography.
Point 1 - Check the possibility of increasing the number of tests to validate the equipment. Why was only the DJ chosen with a 30 cm box? In addition, will the device provide the same results for other highs?
We standardise the height of the box at 30 cm to obtain the best RSI, considering the diversity of the sample. It has been shown that, at 20 centimetres, athletes do not have enough muscle pre-activation time to produce higher heights (1) and this considerably favours contact times (2), so it could affect this parameter (2, 3). Also, measurements from a height of 40 cm may require a certain level of training for inexperienced athletes.
It is important to remember that the RSI test aims to find the ideal height where the athlete can perform at their best (2). During the evaluation of the RSI with different populations, it has been found that the flight time (FT) and the ground contact time (GCT) variation does not present statistically significant differences with the variation of the height of the box (4-6). Therefore, in our opinion, the use of different heights could not be necessary. These criteria help us reinforce the idea that when results from a single height of the box present a large variability in jump heights and contact times (as was the case in our study), this could be considered an ideal evaluation context for the validation of equipment seeking to measure RSI. The 30 cm height could be considered neutral to favour the validations of devices whose primary intention is to assess RSI.
The statistical power of this study reach a Beta value of 86,2%, with an Alpha of 0,05 and a moderate effect size of 0.6 and demanding a sample size of 21 subjects. We add this information in the statistical section.
Point 2 - It is necessary to insert a figure with a photo of the devices, further detailing the experimental protocol.
Thanks to the reviewer's remark. We agree and include photos in the methods section, and we change the description of the experimental protocol according to the reviewer's suggestion.
Point 3 - Explain the exclusion of flat feet subjects.
We thank the reviewer for the opportunity to clarify this exclusion criterion. It is known that flat feet are a prevalent condition that tends to cause pain and alterations in plantar load distribution (7). Moreover, considering that these could be associated with some lower limb injuries (8), we prefer not to submit athletes to high-risk situations.
Point 4 - A more detailed description of the equipment should be performed, especially for Push 2.0.
We thank the reviewer for the remark. We correct the second paragraph related to the Instruments in the methodology section adding the solicited information: Page 3.
Push Band 2.0 (Toronto, Canada) is a device that works with Pushcore, the latest algorithm for using the triaxial accelerometer to provide peak and average velocity data, with a full range of ±16 g and a sensitivity of 2048 least significant bits/g. It incorporates a 3D gyroscope with a full range of ±2000 degrees/s and a sensitivity of 16.4 bits. It measures instantaneous vertical velocity with a sampling rate of 1000 Hz. It has a rechargeable lithium polymer battery with a charge cycle of 1.5 hours with a power consumption of 50 mA during charging. It is 77.5mm high, 55.3mm wide, 15mm deep and weighs 32g. This equipment is compatible with iOS 12 and above, Android 7.0 or above using Bluetooth 4.2 and above transmission protocol. It was installed according to the brand's specifications, and collected data was recorded on a Lenovo yoga tablet and Android 9 operating system. PB2.0 records the height of the jumps using the following previously published equation (9):
Point 5- The equations are not numbered. Explain the principle of equation 1. Where was it based?
We thank the reviewer for this remark that improves the quality of the text. Equations have now been listed in the text.
Equations have been listed in the text.
Equation 1 is based on the flight phase obtained through the studied devices. The flight phase is considered a start point and a finished point, and it is determined your variation through (9)
Jump height=(g×t_flight^2)/8
Where g is the gravitational constant ( and is the flight time obtained through the studied devices. Bosco et al. (1983) described all procedures to determine the jump height.
Point 6 - The authors did not mention with more specificity the advantages of the use PB2.0. It needs to be highlighted.
We thank the reviewer for the opportunity to clarify this point. More content on the advantages was added in the third paragraph of the introduction.
Recently, an accelerometer-based device, the Push Band 2.0 (Push now partner WHOOP, Toronto, Canada), provided, among other measures, jump height and RSI on DJ performance. In addition to its relatively affordable cost, If confirmed, the validity of this device would present numerous advantages for coaches as they spend several weeks far from their athletes along with year-round athletic preparation determined by sports elite level. The validation of this device for jump training would allow RSI metrics to be observed in real-time through the integrated web-based data transmission system. In addition, during exercising tasks at the training, it is difficult to predict with the naked eye when the athlete has a slight decrease in reactive strength, a parameter that this device can detect, indicating to stop the DJ's training immediately...
Point 7 - Figures 1, 2, and 3 have poor quality and two legends. Please, correct them.
Thank the reviewer for the remark. It was corrected in the document.
Point 8- Revise the name of the authors.
Thank the reviewer for the remark. We corrected it in the document.
Point 9- Please insert a space between the number and the unities.
Thank the reviewer for the remark. We corrected it in the document.
Point 10- “Some research [15-17, 19, 20] considers”. If there is more than one author, please reconsider the sentence.
Thank the reviewer for the remark. It was corrected in the document. “Some research [15-17, 19, 20] considered”.
Point 11- Explain the terms after the abstract, such as PF, which is only explained in the abstract.
Thank the reviewer for the remark. We corrected it in the document.
Point 12- “(…) taking, into account (…)”, please, rewrite.
Thank the reviewer for the remark. It was corrected in the document
Point 13- The paragraphs that mention the discussion about Fig. 1 present (A), (B) … (D) to refer to Fig 1a, b, … d. Please, revise all the text to maintain the order in the presentation of the text.
We thank the reviewer and changed the text according to the suggestion.
Point 14- Table 1 title is separated from the table.
We thank the reviewer and changed the text according to the suggestion.
Point 15- “Research [27,28,41] show”, please review the sentence.
We thank the reviewer and changed the text according to the suggestion.
References
- Flanagan EP, Comyns TM. The Use of Contact Time and the Reactive Strength Index to Optimize Fast Stretch-Shortening Cycle Training. Strength & Conditioning Journal. 2008;30(5):32-8.
- Ebben WP, Petushek EJ. Using the reactive strength index modified to evaluate plyometric performance. J Strength Cond Res. 2010;24(8):1983-7.
- Newton RU, Dugan E. Application of strength diagnosis. Strength Cond J. 2002;24(5):50-9.
- Kipp K, Kiely MT, Giordanelli MD, Malloy PJ, Geiser CF. Biomechanical Determinants of the Reactive Strength Index During Drop Jumps. International Journal of Sports Physiology and Performance. 2018;13(1):44-9.
- Bobbert MF, Huijing PA, van Ingen Schenau GJ. Drop jumping. II. The influence of dropping height on the biomechanics of drop jumping. Med Sci Sports Exerc. 1987;19(4):339-46.
- Walsh M, Arampatzis A, Schade F, Brüggemann GP. The effect of drop jump starting height and contact time on power, work performed, and moment of force. J Strength Cond Res. 2004;18(3):561-6.
- Unver B, Erdem EU, Akbas E. Effects of Short-Foot Exercises on Foot Posture, Pain, Disability, and Plantar Pressure in Pes Planus. J Sport Rehabil. 2020;29(4):436-40.
- Buldt AK, Forghany S, Landorf KB, Levinger P, Murley GS, Menz HB. Foot posture is associated with plantar pressure during gait: A comparison of normal, planus and cavus feet. Gait Posture. 2018;62:235-40.
- Bosco C, Luhtanen P, Komi PV. A simple method for measurement of mechanical power in jumping. Eur J Appl Physiol Occup Physiol. 1983;50(2):273-82.
Reviewer 2 Report
The purpose of this study was to evaluate the validity of the PB2.0 device for the Reactive Strength Index (RSI) assessment in drop jumping. Measurement and evaluation are essential items in sports training or sports coaching. Assessing the validity of the measurement devices and the consistency between each device supports the diversity and compatibility of measurement methods. The results of this study have significant implications for measurement and assessment items in sports training and sports coaching.
In my opinion, there is a clear rationale for the statistical treatment methods, and the methodology has been well explored. I do not see any statistical problems.
Major Comments
In this study, 20 subjects were measured a defined number of times and in a defined number of disciplines. I think the sample size should be considered and clearly stated in advance of the study to assess its validity of the study. I believe that a clear statement regarding the rationale for the sample size is necessary.
In this study, a drop jump of 30 cm (30 DJ) was employed as the measurement. Concerning the study's limitations, the author states that it would be interesting to see other height settings. I believe that clarifying the rationale for adopting the 30DJ category would describe the purpose of the study.
In Table 1, there is a point of clarification regarding the notation of height. For example, the standard deviation of the data for the 12 male subjects is 0.06. Perhaps there is an inconsistency in units. Please reconfirm. If there is no error, it is my lack of recognition. Please forgive me.
Author Response
Response to Reviewer 2 Comments.
Thank you very much for your interesting comments and contributions to the improvement of the study. The authors appreciate the value of the comments and recognise that they were important in consolidating the document. They will also try to respond to each question based on a review of the content supported by the previously published bibliography.
Point 1: In this study, 20 subjects were measured a defined number of times and in a defined number of disciplines. I think the sample size should be considered and clearly stated in advance of the study to assess its validity of the study. I believe that a clear statement regarding the rationale for the sample size is necessary.
The a priori sample size calculation estimates a statistical power reaching a Beta value of 86,2%, with an Alpha of 0,05 and a moderate effect size of 0.6, demanding a sample size of 21 subjects. We add this information in the statistical section.
Additionally, this sample realised a total of 60 trials, where the RSI and height jump measurements were considered. The highly competitive level of the athletes should be considered due to the higher stability in the jump performance.
The mean difference between the RSI acceded through the Pushband 2.0 and that from the force plate showed a trivial (0.2) Cohen’s d effect size
Point 2: In this study, a drop jump of 30 cm (30 DJ) was employed as the measurement. Concerning the study's limitations, the author states that it would be interesting to see other height settings. I believe that clarifying the rationale for adopting the 30DJ category would describe the purpose of the study.
We thank the reviewer for this remark. This sentence was withdrawn as the assessment at different heights should not be considered a limitation of this criterion.
The RSI has a fundamental objective to find the ideal height where the athlete can manifest his best performance (1). During the evaluation of the RSI with different populations, it has been found that the FT and GCT variation does not present statistically significant differences with the height of the box (2-4). Therefore, it might not make sense to perform validations at different heights to find different results. These criteria help us to reinforce the idea that when sample results for a single height present a large variability in jump heights and contact times (as was the case in our study), this could be considered an ideal evaluation context for the validation of equipment seeking to measure RSI. The 30 cm height could be considered neutral to the validation process of devices whose primary intention is to assess RSI.
Point 3: In Table 1, there is a point of clarification regarding the notation of height. For example, the standard deviation of the data for the 12 male subjects is 0.06. Perhaps there is an inconsistency in units. Please reconfirm. If there is no error, it is my lack of recognition. Please forgive me.
We thank the reviewer for the comment. It has been corrected in the text.
References
- Ebben WP, Petushek EJ. Using the reactive strength index modified to evaluate plyometric performance. J Strength Cond Res. 2010;24(8):1983-7.
- Kipp K, Kiely MT, Giordanelli MD, Malloy PJ, Geiser CF. Biomechanical Determinants of the Reactive Strength Index During Drop Jumps. International Journal of Sports Physiology and Performance. 2018;13(1):44-9.
- Bobbert MF, Huijing PA, van Ingen Schenau GJ. Drop jumping. II. The influence of dropping height on the biomechanics of drop jumping. Med Sci Sports Exerc. 1987;19(4):339-46.
- Walsh M, Arampatzis A, Schade F, Brüggemann GP. The effect of drop jump starting height and contact time on power, work performed, and moment of force. J Strength Cond Res. 2004;18(3):561-6.
Reviewer 3 Report
Comments to the Author:
I thank to the editors for the opportunity to review this study. The present manuscript by Montoro et al., analyzed “The validity of the Push Band 2.0 on the Reactive Strength Index assessment in Drop jump”. The authors with this study try to evaluate the validity of the PB2.0 device for RSI assessment in the drop jump, using a force plate and optical measurement system as standard references. Overall, the study addresses a topic that could be interesting but there are numerous problems and issues that need to be addressed.
General Comments:
1. There is an error in the names of the authors.
2. I recommend authors to redo their introduction with a maximum of three or four paragraphs. Authors in their introduction uses 6 small paragraphs that make the reader lose the thread of the story you want to convey. Focus your introduction and 3 or 4 paragraphs well-formed and put together and it has a common thread that leads the reader to the objective of the study. An introduction with different paragraphs confuses the reader.
3. The authors should consider justifying in some way that 20 subjects are enough to validate the section to be evaluated, in fact I doubt that it can be so. I need to appreciate the statistical power of the sample to be used.
4. The code of ethics committee that endorses the study must be added. It is not sufficient to simply add the name of the institution.
5. BMI (kg.m-2) correct this error.
6. The figures are not of sufficient quality, the authors should change the figures and add more quality so that the information they want to give can be appreciated perfectly. In addition, it should be added what each letter of the figures represents, i.e., A, B, C and D.
7. What were the strengths of this study?
Author Response
Response to Reviewer 3 Comments.
We thank the reviewer's critical comments and contributions to the improvement of the paper. The authors appreciate the value of the remarks and recognise that they were important in consolidating the document. They will also try to respond to each question based on a review of the content supported by the previously published bibliography.
- There is an error in the names of the authors.
We thank the reviewer for the comment, which has been corrected in the text.
- I recommend authors to redo their introduction with a maximum of three or four paragraphs. Authors in their introduction uses 6 small paragraphs that make the reader lose the thread of the story you want to convey. Focus your introduction and 3 or 4 paragraphs well-formed and put together and it has a common thread that leads the reader to the objective of the study. An introduction with different paragraphs confuses the reader.
We thank the reviewer for the suggestion. We try to build an introduction that follows this rationale:
- Highlight the plyometric exercises, the Reactive Strength Index, the existence of the device, and the importance of validation.
- For this reason, we revise the introduction, cleaning some superfluous topics and adjusting the alignment of the main idea. We hope we have satisfied the reviewer's point of view.
- The authors should consider justifying in some way that 20 subjects are enough to validate the section to be evaluated, in fact I doubt that it can be so. I need to appreciate the statistical power of the sample to be used.
The a priori sample size calculation estimates a statistical power reaching a Beta value of 86,2%, with an Alpha of 0,05 and a moderate effect size of 0.6, demanding a sample size of 21 subjects. We add this information in the statistical section.
Additionally, this sample realised a total of 60 trials, where the RSI and height jump measurements were considered. The highly competitive level of the athletes should be considered due to the higher stability in the jump performance.
The mean difference between the RSI acceded through the Pushband 2.0, and that from the force plate showed a trivial (0.2) Cohen's d effect size
- The code of ethics committee that endorses the study must be added. It is not sufficient to simply add the name of the institution.
We thank the reviewer’s comment. However, this information was located at the end of the document, under “Funding”
Institutional Review Board Statement: The study was conducted in accordance with the Declaration of Helsinki, and approved by the Ethics Committee of Coimbra University (protocol code: CE/FCDEF-UC/00802021 the 6 July 2021)." for studies involving humans.
- BMI (kg.m-2) correct this error.
We thank the reviewer's remark. It has been corrected in the text.
- The figures are not of sufficient quality, the authors should change the figures and add more quality so that the information they want to give can be appreciated perfectly. In addition, it should be added what each letter of the figures represents, i.e., A, B, C and D.
We thank the reviewer's comment. The graphics have been changed to higher quality ones. On the other hand, the authors revised paragraphs by modifying the presentation and the correspondence of each letter.
Colocar como decidimos ontem (Fig..a, etc)
- Fig3a represents the mean difference RSI values between PF and PB2.0…
- Fig 3b represents the mean difference values of RSI of OPT…
- Fig 3c represents the mean difference of the RSI between PF and OPT …
- Fig 3d the mean difference of the HJ between PF and OPT...
What were the strengths of this study?
The recognition that the Drop Jump is a powerful mean of plyometrics justify its monitoring. The possibility of using a portable and affordable device for this purpose would be of interest to the sports community. The validation of this device for jump training would allow RSI metrics to be observed in real-time through the data transmission system integrated into the web. In addition, during training sessions, when athletes perform jump tasks, it is difficult to predict with the naked eye when the athlete has a slight decrease in reactive strength, an element that the device can alert to stop the DJ's training immediately.
Another element to consider that represents a small contribution is the importance of correctly fixing the device to the waist and inside the holster. Since the DJ is an exercise that requires a rapid change of movement, the slightest displacement of the device could lead to errors in the measurement. This last criterion is not clearly stated in the manufacturer's specifications.
Round 2
Reviewer 1 Report
The paper presented improvements in relation to the old version, however, I still highlight some corrections to be made.
“In addition to its relatively affordable cost, If confirmed, the validity of this device would present numerous advantages for coaches as they spend several weeks far from their athletes along with year-round athletic preparation deter-mined by sports elite level.”. Please, correct the sentence.
“Push Band 2.0 (Figure 2) Toronto, Canada is a device that works with Pushcore,”. Please, correct the sentence
Equations (1) and (2) are out of the journal’s template.
Figure 1, 2, 4, and 5 legends are out of the template.
“Figure 4. presents the agreement between the used methods.” Correct the sentence of the legend.
Author Response
We thank the reviewer for the comments. The authors consider that they have been reviewed and corrected in the attached document.
Reviewer 3 Report
To the author:
I appreciate the effort of the authors. It is evident that the authors have spent considerable time revising the manuscript and their hard work is clearly paying off. This manuscript is drastically improved over the original submission. The message is very clear, the language is much cleaner, and the problems of the first version have been corrected. In addition, the authors have responded satisfactorily to all my comments. Finally, I recommend the authors to remove this sentence "We add this information in the statistics section" in the methods section, as it is misleading. Once this issue is corrected, I encourage the editor to consider this manuscript for publication because of the interesting value of the study conducted, which is now a much more robust study.
Author Response
Thank the reviewer for the remark. It was corrected in the document